# Rate of Torque Development in the Quadriceps after Anterior Cruciate Ligament Reconstruction with Hamstring Tendon Autografts in Young Female Athletes

**DOI:** 10.3390/ijerph191811761

**Published:** 2022-09-18

**Authors:** Makoto Suzuki, Tomoya Ishida, Mina Samukawa, Hisashi Matsumoto, Yu Ito, Yoshimitsu Aoki, Harukazu Tohyama

**Affiliations:** 1Faculty of Health Sciences, Hokkaido University, Sapporo 060-0812, Japan; 2Department of Rehabilitation, Hokushin Orthopaedic Hospital, Sapporo 060-0908, Japan; 3Department of Orthopaedic Surgery, Hokushin Orthopaedic Hospital, Sapporo 060-0908, Japan

**Keywords:** anterior cruciate ligament reconstruction, quadriceps function, patient-reported knee function

## Abstract

This study aims to compare the limb symmetry index (LSI) of the rate of torque development (RTD) of the quadriceps with that of the peak torque after anterior cruciate ligament reconstruction (ACLR) using semitendinosus and gracilis tendon (STG) autografts and to investigate the associations of the LSI of torque parameters with patient-reported knee function. The participants included 23 female athletes after ACLR with STG grafts. Isometric quadriceps tests were performed using an isokinetic dynamometer. The peak torque, RTD_100_ (0 to 100 ms) and RTD_200_ (100 to 200 ms) were determined using torque-time curves. Comparisons of the LSI of torque parameters was performed by ANOVA. Univariate regression analysis was used to examine the relationship between the LSI of torque parameters and the patient-reported knee function. The LSIs of the peak torque and RTD_200_ were significantly lower than that of the RTD_100_ (*p* = 0.049, *p* = 0.039, respectively). Regression analysis showed that the LSI of the peak torque was associated with the patient-reported knee function (*R*^2^ = 0.40, *p* = 0.001). It would be useful to evaluate the peak torque in young female athletes under the age of 18 and at 8–10 months after ACLR with STG grafts.

## 1. Introduction

Anterior cruciate ligament (ACL) injury is a serious sports trauma and accounts for 25–50% of all sports-related injuries to the knee [1,2]. Although ACL reconstruction (ACLR) can restore joint stability [3,4], many problems have been reported after ACLR, such as secondary ACL injury, low subjective knee function, and high risk of posttraumatic osteoarthritis [5,6,7]. A meta-analysis reported that the incidence of secondary ACL injury after returning to sports was 23% [6]. The International Knee Documentation Committee Subjective Knee Evaluation Form (IKDC-SKF), known as a patient-reported outcome measure, revealed lower subjective knee function in ACLR patients at even 4 years after ACLR compared to that in uninjured individuals [5]. A systematic review on posttraumatic osteoarthritis after ACL injury showed that 30–50% of patients had radiographic signs of osteoarthritis 12 years after ACLR [7].

Quadriceps femoris dysfunction after ACLR is a significant problem [8] and is associated with secondary ACL injury, low subjective knee scores, and osteoarthritic changes after ACLR [9,10,11,12,13]. Previous studies reported that quadriceps strength after ACLR was associated with interlimb asymmetry during landing [9,10,11], which was one of the risk factors for secondary ACL injury [14]. In addition, quadriceps dysfunction after ACLR affected the low subjective knee function and tibiofemoral joint space width narrowing [12,13]. Therefore, clinicians should consider the importance of fully restoring quadriceps strength after ACLR and postoperative rehabilitation.

The rate of torque development (RTD) of the quadriceps has been proposed as an index to evaluate quadriceps strength after ACLR [15,16,17,18,19,20]. The RTD is force production per unit time, which reflects the capacity for instantaneous force production and development [21]. The limb symmetry index (LSI) of the quadriceps peak torque was reported to be approximately 80%, while that of the RTD was only 50–70% at 6 months after ACLR with bone-patellar-tendon bone (BTB) grafts [17,18]. These results indicated that the RTD may be better for determining the interlimb difference in quadriceps strength than the peak torque after ACLR with BTB grafts. The RTD within 100 msec after torque production is affected by the neural drive and firing motor unit frequency [22]. On the other hand, the RTD after 100 msec is affected by the stiffness of the muscle-tendon complex and the peak torque production potential [22,23]. In animal models, removal of the central third of the patellar tendon decreased the stiffness of the patellar tendon [24,25]. Thus, the RTD after ACLR with BTB grafts may be affected by changes in the stiffness of the patellar tendon. In fact, some studies showed that both the LSI of the quadriceps peak torque and RTD were 80–90% after ACLR with autografts, including semitendinosus and gracilis tendon (STG) in addition to BTB grafts [15,19]. Therefore, the LSI of the RTD after ACLR with STG grafts may not be much different from that of peak torque unlike ACLR with BTB grafts. However, there is no report comparing the LSI of peak torque and RTD after ACLR with STG grafts. This comparison will show whether the peak torque or the RTD is better able to determine the interlimb difference in quadriceps strength after ACLR with STG grafts. The present study aimed to determine the interlimb difference in the RTD and peak torque and to compare the LSI of the RTD with that of the peak torque after ACLR with STG grafts. Furthermore, it is unclear whether the LSI of the RTD or peak torque predicts patients-reported knee function. Thus, the second purpose of this study was to investigate the associations of the LSI of the RTD or peak torque on the IKDC-SKF score. We hypothesized that interlimb difference would remain in the RTD within 100 ms and 200 ms, and peak torque, whereas there would be no differences among the LSIs of these parameters. Furthermore, we hypothesized that the LSI of the RTD and peak torque would predict the IKDC-SKF score.

## 2. Materials and Methods

### 2.1. Participants

The present study was conducted with a cross-sectional design. Twenty-three female athletes after ACLR with STG tendon autografts [26,27] participated in the present study (age: 16.8 ± 1.2 years; height: 160.3 ± 4.7 cm; body weight: 57.3 ± 6.4 kg; time from surgery: 8.7 ± 0.9 months; preinjury modified Tegner activity scale scores: 7.2 ± 0.5). The inclusion criteria were as follows: age younger than 18 years, unilateral ACL injury and modified Tegner activity scale scores ≥ 7 prior to ACL injury. The sports of the participants were basketball (*n* = 13), volleyball (*n* = 7), soccer (*n* = 1), badminton (*n* = 1), and judo (*n* = 1). Since it was reported that age and sex affect the recovery of quadriceps strength after ACLR, the present study was limited to age and sex [28,29]. All participants with a history of any orthopedic surgery other than ACLR or neurological disorders of the lower limb were excluded from the study. Patients with concomitant grade II or III posterior cruciate ligament or medial collateral ligament injuries and those with International Cartilage Research Society (ICRS) grade 2 or higher cartilage injury were also excluded. Participants with meniscus injuries were included. The time from injury to surgery was 2.7 ± 3.2 months. Six participants underwent concomitant meniscus repair. Of the 23 participants, 14 participants had ACLR on the non-dominant leg and 9 participants had ACLR on the dominant leg.

All participants completed a standardized rehabilitation protocol. The participants were allowed to start running at 12 weeks and to jump and sprint with submaximal effort from 5 months. They were allowed to return to sports between 8 and 10 months after surgery. The present study was approved by the institutional review board of the authors’ affiliated institution (approval number: 18-64). All the participants and their guardians received a written document explaining the study objectives and procedures and were required to provide written informed consent before participating in the research activities.

### 2.2. RTD and LSI

Each participant performed a 5-min warm-up using a stationary bike at self-selected speeds [30]. After the warm-up, each participant performed three practice sets of 5 s maximal voluntary isometric contraction (MVIC) of the quadriceps before the actual measurement. The knee extension torque was recorded using an isokinetic dynamometer (Biodex System 3, Biodex Medical Systems, Inc., Shirley, NY, USA). The sampling rate was set at 100 Hz. All trials were performed with the hip at 90° and knee at 70° of flexion [19,21,30]. Straps were firmly fastened around the patient’s chest, waist, and distal thigh for stabilization [16]. A shin pad was placed two finger widths above the lateral malleolus [16]. Before the test, the participant was instructed to extend the knee “as fast and forcefully as possible” and practiced for familiarization using visual feedback from the Biodex monitor screen [30]. The participant then performed each task three times for each leg (uninvolved limb first) with a 1-min rest interval [30]. Verbal encouragement was given to the participants to maximize torque production during the tests.

Custom MATLAB code (The Math Works, Inc., Natick, MA, USA) was used for data processing. The force–time signal was low-pass filtered at 6 Hz using a second-order Butterworth filter [16]. The knee extension torque was measured three times, and the RTD analysis was performed on the trial with the highest torque value. The peak torque was calculated from the maximum value of torque for 3 s after the onset of torque generation (Figure 1). When multiple peaks were observed, the maximum value of the trial was used as the peak torque. The first peak occurred after 200 ms in all participants. The onset of torque generation was defined as the time when the knee extension torque exceeded the baseline by 7.5 Nm [21,22]. The RTD_100_ and RTD_200_ were calculated from the torque-time curves (Figure 1) [22]. Both the peak torque and RTD were normalized to the participant’s height and body weight. The LSI of the peak torque and RTD was calculated as the percentage in the involved limb compared with that in the uninvolved limb as follows:$$LSI\;=\;\left(\frac{involved\;limb}{\;uninvolved\;limb}\right)\times 100\;\left[\%\right]$$

### 2.3. IKDC-SKF

Patients-reported knee function was evaluated using the IKDC-SKF, which is an index comprising a score from 0 to 100 and scores for knee symptoms, function, and sports activities. The IKDC-SKF score is highly relevant and reliable [31], and the Japanese version of the IKDC-SKF used in the study has been reported to have high validity and reliability [32].

### 2.4. Statistical Analysis

Statistical analyses were performed using IBM SPSS Statistics 22 software (IBM, Chicago, IL, USA). First, a Shapiro–Wilk test was performed to confirm the normality of variables, and normality was confirmed for all variables. The paired *t* test was used to examine the between-limb differences in the RTD_100_, RTD_200_ and peak torque. One-way analysis of variance (ANOVA) was used to examine the difference between the LSIs of the RTD_100_, RTD_200_ and peak torque. Post hoc comparisons were conducted using Bonferroni’s test. Furthermore, the value of *d_z_* was calculated as the effect size [33]. A *d_z_* value greater than 0.80 was interpreted as large, 0.50 to 0.79 as moderate, and 0.20 to 0.49 as small [33]. Univariate regression analysis was conducted to examine the association of the IKDC-SKF score with the LSIs of the RTD_100_, RTD_200_, and peak torque. The statistical significance level was set at *p* < 0.05. The sample size was calculated using G*Power 3.1 based on previously published data [34]. More than 18 participants were required to detect any between-limb difference in the RTD_100_ (80% power; α = 0.05; *d_z_* = 0.70).

## 3. Results

The peak torque was significantly lower in the involved limb than in the uninvolved limb, with a large effect size (*p* < 0.001; *d_z_* = 1.000) (Table 1). The RTD_200_ was also significantly lower in the involved limb than in the uninvolved limb, with a large effect size (*p* < 0.001; *d_z_* = 0.830) (Table 1). However, no differences were found in the RTD_100_ between the limbs (Table 1).

One-way ANOVA revealed a significant difference in the LSI of the RTDs and peak torque (*p* = 0.009). The LSI of the peak torque was significantly lower than that of the RTD_100_, as shown by the moderate effect size (*p* = 0.049; *d_z_* = 0.541) (Figure 2). The LSI of the RTD_200_ was also significantly lower than that of the RTD_100_, with a moderate effect size (*p* = 0.039; *d_z_* = 0.563) (Figure 2). There was no significant difference in the LSI of the RTD_200_.

The mean IKDC-SKF score was 87.4 ± 11.0. Univariate regression analysis revealed that the LSI of the peak torque predicted the IKDC-SKF score *(R*^2^ = 0.40; *β* = 0.63; *p* = 0.001), while the LSIs of the RTD_100_ (*R*^2^ = 0.14; *β* = 0.38; *p* = 0.074) and RTD_200_ were not predicted (*R*^2^ = 0.11; *β* = 0.33; *p* = 0.126) (Figure 3).

## 4. Discussion

We aimed to determine the interlimb difference in the RTD and peak torque and to compare the LSI of the RTD with that of the peak torque after ACLR with STG grafts. Our findings showed that the RTD_200_ and the peak torque were significantly lower in the involved limb than in the uninvolved limb and that the LSIs of the peak torque and RTD_200_ were significantly lower than that of the RTD_100_. Additionally, we investigated the associations of the IKDC-SKF score with the LSI of the RTD and peak torque. The LSI of the peak torque was the only variable that could predict the IKDC-SKF. These results partially supported our hypothesis.

The RTD_200_ and peak torque were significantly lower in the involved limb than in the uninvolved limb. No difference was found between the LSI of RTD_200_ and the LSI of the peak torque. These findings suggest that the LSI of the RTD_200_ is not different from the LSI of the peak torque in detecting quadriceps weakness after ACLR with STG grafts. The RTD_200_ was affected by structural factors in relation to musculotendinous stiffness [23]. In animal studies, the recovery of patellar tendon stiffness required approximately 1 year [35,36]. The LSI of the RTD_200_ at 6 months after ACLR with BTB grafts was 43% [17]; even at 4 years after ACLR with BTB or STG grafts, it was up to 78% [15]. In our study, the LSI of the RTD_200_ was 87.6%, which is higher than that reported previously [15,17]. Although a direct comparison between the previous studies and the present study is difficult due to the different age groups, genders, and postoperative periods, the RTD_200_ of the quadriceps after ACLR with STG grafts is less likely to decrease than that after ACLR with BTB grafts. Additional longitudinal studies should be conducted to clarify the effects of the graft type on RTD_200_ recovery after ACLR. Peak torque is most used to evaluate the quadriceps strength after ACLR, and there are many reports on its usefulness [9,10,11,13]. The LSI of the peak torque in this study was 86.7%, which suggests that interlimb asymmetry can be detected as in a previous study [15,19]. Therefore, the RTD_200_ and peak torque would be useful as indicators to detect quadriceps asymmetry in young female athletes at 8–10 months after ACLR with STG grafts.

The LSI of RTD_100_ was significantly higher than those of RTD_200_ and peak torque. No significant difference was found in the RTD_100_ between the involved and uninvolved limbs. These results indicate that the RTD_100_ of the involved limb is comparable to that of the uninvolved limb, which did not support our hypothesis. The recovery of neural function after ACLR manifests differently depending on the type of tendon graft. A recent systematic review revealed that neural drive of the quadriceps in the involved limb after ACLR with BTB grafts was lower than that in the uninvolved limb [37]. In contrast, the neural drive of the quadriceps in the involved limb after ACLR with STG grafts was higher than that in the uninvolved limb [37], which supported the present findings. Thus, this study result showed that female young athletes had good RTD_100_ recovery at 8–10 months after ACLR with STG grafts.

Univariate regression analysis revealed that the LSI of the peak torque predicted the IKDC-SKF score. The results of the present study partially support our hypothesis that the LSIs of the RTD and peak torque would predict the IKDC-SKF score, which is consistent with previous findings showing that the IKDC-SKF score was significantly correlated with the LSI of the peak torque [12,38]. A previous study also showed no significant correlation between the LSI of RTD and IKDC-SKF at 4 years after ACLR [15], while another study showed a significant correlation between them at 3 months after ACLR [16]. Therefore, the association between the IKDC-SKF score and the LSI of the RTD would be observed only in the early postoperative period. The results of the present study show that the LSI of the peak torque is a more associating outcome with the IKDC-SKF compared to the LSI of the RTD in female athletes at 8–10 months after ACLR with STG grafts. Therefore, the peak torque assessment would be important within the period.

Concerning clinical application, RTD_100_, RTD_200_, and peak torque were used as assessments of quadriceps strength after ACLR [9,10,11,15,16,17,18,19,20]. However, there have been no reports on which parameters are more sensitive for detecting interlimb asymmetry. The findings of the present study showed that the RTD_200_ and peak torque were more sensitive for detecting interlimb asymmetry after ACLR with STG grafts. Moreover, only the LSI of peak torque showed a significant correlation with the IKDC-SKF score. Previous studies also reported that the asymmetry in the peak torque after ACLR is associated with the interlimb asymmetry of knee kinematics and kinetics during landing [9,10,11]. Therefore, the peak torque would be a more useful assessment of quadriceps strength than the RTD after ACLR with STG grafts. A previous study reported that the LSI of RTD was lower than that of peak torque at 6 months after ACLR with BTB grafts [18]. The present findings suggest that neither peak torque nor RTD are always the best assessment method after ACLR. The study findings suggest that the peak torque rather than RTD is recommended for evaluation at the return to sports period after ACLR with STG. Furthermore, rehabilitation exercises to increase muscle volume is necessary during this phase. Hence, the RTD assessment and the activation of neural system should be important after ACLR with BTB [18]. Further longitudinal studies should clarify to see the differences in RTD and the peak torque recovery course between the grafts to optimize quadriceps training after ACLR.

Some limitations of this study should be addressed. First, the participants were limited to young female athletes only. Age and sex differences may affect quadriceps strength [28,29]. Second, ACLR of all participants was performed with STG grafts. The results may differ according to the type of ACLR graft—namely allograft, quadriceps tendon graft, or BTB graft. Third, it was a clinical setting study conducted in a single hospital. As such, different rehabilitation protocols may lead to different results. Fourth, we did not measure multiple time points. The present study results were obtained from participants 8–10 months after ACLR, and the results may differ at other time points. Finally, knee function was assessed based on the subjective knee score only and no functional dynamic tasks were determined in this study.

## 5. Conclusions

The present study shows that the RTD_200_ and peak torque were more sensitive for detecting interlimb asymmetry after ACLR with STG grafts. The LSI of peak torque showed a significant correlation with the IKDC-SKF score, but RTD did not show any significant correlation. These findings indicate that peak torque may be a more useful assessment of quadriceps strength after ACLR with STG grafts than RTD. Moreover, training to enlarge quadriceps muscle volume to increase peak torque is recommended over activation of the neural system close to returning to sporting activities.

## Figures and Tables

**Figure 1 ijerph-19-11761-f001:**
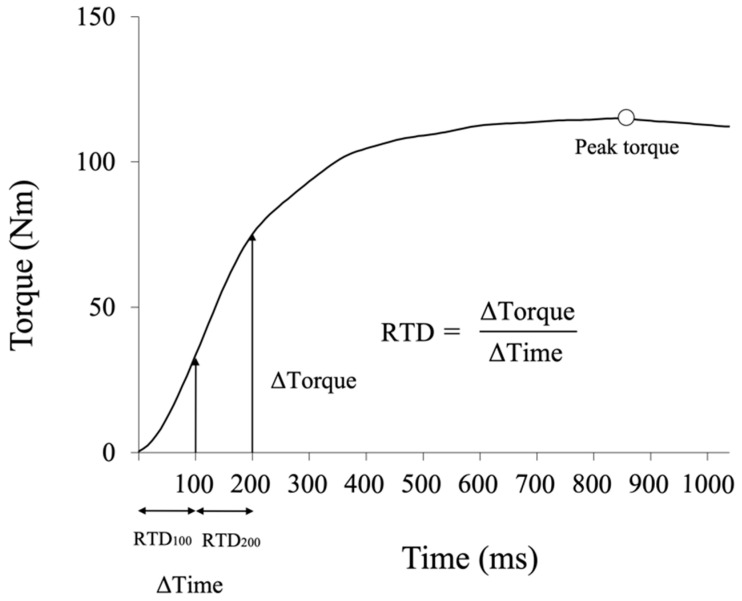
Rate of torque development (RTD) from the slope of the torque–time curve. RTD_100_ is calculated from 0 to 100 ms, and RTD_200_ is calculated from 100 to 200 ms.

**Figure 2 ijerph-19-11761-f002:**
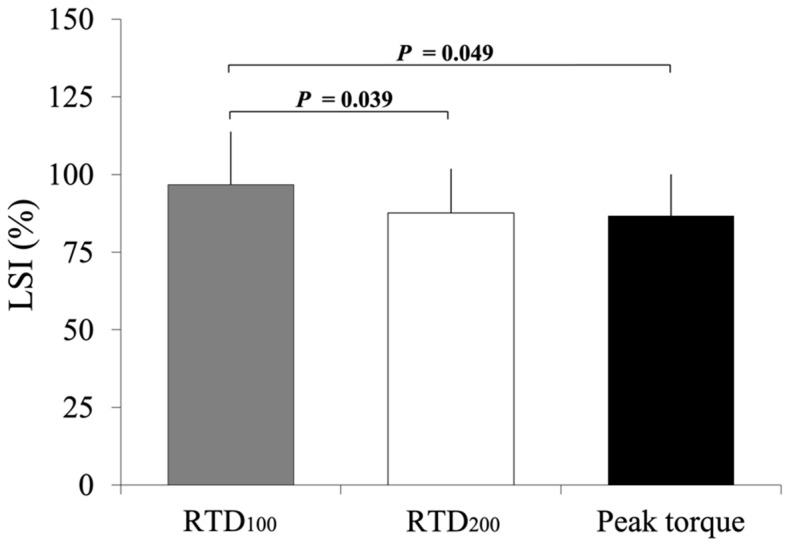
LSI of RTD_100_, RTD_200_ and peak torque. *p*-values indicate the results of post hoc tests. RTD, rate of torque development.

**Figure 3 ijerph-19-11761-f003:**
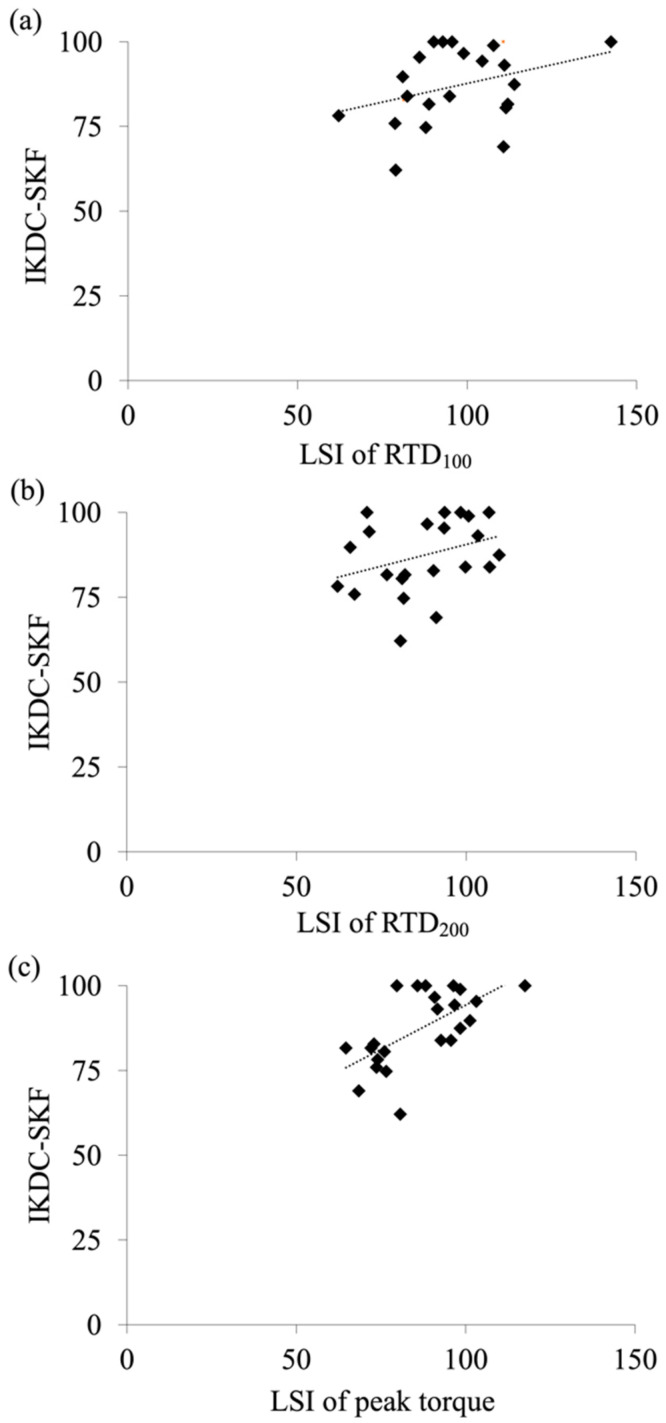
Association between IKDC-SKF score and the LSI of the torque parameters. (**a**) RTD_100_, (**b**) RTD_200,_ (**c**) Peak torque.

**Table 1 ijerph-19-11761-t001:** Interlimb differences in the RTD_100_, RTD_200_, and peak torque.

	Involved Limb	Uninvolved Limb	LSI	*p*-Value *	*d_z_*
RTD_100_, Nm/s/kg/m	5.9 (1.9)	6.1 (1.8)	96.7 (17.1)	0.304	0.219
RTD_200_, Nm/s/kg/m	4.1 (1.3)	4.7 (1.2)	87.6 (14.2)	<0.001	0.830
Peak torque, Nm/kg/m	1.5 (0.3)	1.8 (0.3)	86.7 (13.3)	<0.001	1.000

Mean (SD). * indicates the results of a paired *t*-test comparing the involved limb with the uninvolved limb. Significant differences are shown in bold. RTD_100_, rate of torque development from 0 to 100 ms; RTD_200_, rate of torque development from 100 to 200 ms.

## Data Availability

The datasets in this study are available upon reasonable request to the corresponding author’s e-mail.

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
