# Peer review of "Rate of Torque Development in the Quadriceps after Anterior Cruciate Ligament Reconstruction with Hamstring Tendon Autografts in Young Female Athletes"

_ijerph, 2022, doi:10.3390/ijerph191811761_

Round 1
Reviewer 1 Report
Dear Authors
First of all, thank you for submitting to IJERPH. Overall, this article is well written.
This study was conducted by linking RTD and LSI. This is an interesting and meaningful study. However, I hope it will be improved a bit for publication.
If possible, I would suggest replacing the references a bit with those from the last 5 years.
I suggest that the section be as follows:
2.2 procedures => delete and replace “2.2 RTD and LSI”, “2.3 IKDC-SKF”, “2.4 Statistical analysis”
Line 121: 2.3. Data analysis include to “2.2 RTD and LSI”
Add a significance level to your “statistical analysis”. Is p < 0.05 ?
Change the order of the tables and figures to the following for readability.
Peak torque, RTD100, RTD200 (X)
RTD100, RTD200, Peak torque (O)
This study has many limitations.
Is there a difference between autograft and allograft?
Is there any difference from using other tendons?
Is there any difference according to the post-operative management method (eg, rehabilitation method)?
Multiple time points were not measured.
Therefore, I request that the following be added.
How will the results of this study be used and applied in the field or clinical?
What does it mean?
How should the therapist use these results to help the patient?
Or, what services or understanding should the therapist provide based on the results of this study?
Conclusion
The conclusion is the level at which the results are repeated. A more clear, additional message, please.
Author Response
We thank you for carefully reading our manuscript and for providing helpful comments. We have revised the manuscript according to your valuable comments. Please refer to the following point-by-point responses. Changes made in the manuscript are indicated in the attached file.

Reviewer 2 Report
Deatr authors,
Your paper seems of scientific interest but some concerns have to be addressed.
First of all, in the method section, You have to better clarify what is the study design model that You followed, so as to be able to justify the statistical analysis You carried out.
Then, the discussion section must be improved. In this section, it is important to explain why Your findings are useful for new researches in the field. Particularly, it is desirable to highlight the importance of Your results from a rehabilitative point of view. Despite the small sample size, You could compare Your interesting findings with other studies regarding sport rehabilitation and other rehabilitation approaches. To do that, I suggest to use the following references:
-Farì G, Santagati D, Macchiarola D, Ricci V, Di Paolo S, Caforio L, Invernizzi M,
Notarnicola A, Megna M, Ranieri M. Musculoskeletal pain related to surfing practice: Which role for sports rehabilitation strategies? A cross-sectional study. J Back Musculoskelet Rehabil. 2022;35(4):911-917. doi: 10.3233/BMR-210191. PMID: 35068441.
-Notarnicola A, Maccagnano G, Farì G, Bianchi FP, Moretti L, Covelli I, Ribatti P, Mennuni C, Tafuri S, Pesce V, Moretti B. Extracorporeal shockwave therapy for plantar fasciitis and gastrocnemius muscle: effectiveness of a combined treatment. J Biol Regul Homeost Agents. 2020 Jan-Feb;34(1):285-290. doi: 10.23812/19-347-L. PMID: 32191019.
- Latino F, Cataldi S, Bonavolontà V, Carvutto R, De Candia M, Fischetti F. The Influence of Physical Education on Self-Efficacy in Overweight Schoolgirls: A 12-Week Training Program. Front Psychol. 2021 Nov 3;12:693244. doi: 10.3389/fpsyg.2021.693244. PMID: 34803792; PMCID: PMC8595474.
Best regards and good luck
Author Response

(The authors gave the same response as above.)

Reviewer 3 Report
Dear Authors,
First of all, I think that the manuscript entitled: " Rate of torque development in the quadriceps after anterior cruciate ligament reconstruction with hamstring tendon auto-grafts in young female athletes" submitted for publication in the International Journal of Environmental Research and Public Health, has both scientific and clinical interest.
Comments:
* Lines 81 - 84:
I think it would be useful for the authors to add the training age of the subjects as well as their specific sport.
Additionally, in case the researchers had administered lower extremity lateralization (footedness) questionnaire to the subjects of the study, it would be useful to report how many of the subjects had suffered an ACL tear in the dominant lower extremity and how many in the non-dominant extremity. In my opinion, this is particularly important to mention, compared to the results of the study in the discussion section of this article.
Author Response

(The authors gave the same response as above.)
